# Molecular Epidemiology and Presence of Hybrid Pathogenic *Escherichia coli* among Isolates from Community-Acquired Urinary Tract Infection

**DOI:** 10.3390/microorganisms10020302

**Published:** 2022-01-27

**Authors:** Júllia A. S. Nascimento, Fernanda F. Santos, José F. Santos-Neto, Liana O. Trovão, Tiago B. Valiatti, Isabel C. Pinaffi, Mônica A. M. Vieira, Rosa M. Silva, Ivan N. Falsetti, Ana C. M. Santos, Tânia A. T. Gomes

**Affiliations:** 1Laboratório Experimental de Patogenicidade de Enterobactérias (LEPE), Disciplina de Microbiologia, Departamento de Microbiologia, Imunologia e Parasitologia (DMIP), Escola Paulista de Medicina (EPM), Universidade Federal de São Paulo (UNIFESP), São Paulo 04023-062, Brazil; jullia.nascimento@unifesp.br (J.A.S.N.); jfs.neto@unifesp.br (J.F.S.-N.); lianatrovao@hotmail.com (L.O.T.); mvmonicavieira@gmail.com (M.A.M.V.); carolina.mello@unifesp.br (A.C.M.S.); 2Laboratório Alerta, Disciplina de Infectologia, Departamento de Medicina, Escola Paulista de Medicina (EPM), Universidade Federal de São Paulo (UNIFESP), São Paulo 04039-032, Brazil; ff.santos@unifesp.br (F.F.S.); tiago.valiatti@unifesp.br (T.B.V.); 3Laboratório Santa Cruz Medicina Diagnóstica, Mogi Guaçu 13840-052, Brazil; isabelpinaffi@labsantacruz.com (I.C.P.); ivan.falsetti@gmail.com (I.N.F.); 4Laboratório de Enterobactérias, Disciplina de Microbiologia, Departamento de Microbiologia, Imunologia e Parasitologia (DMIP), Escola Paulista de Medicina (EPM), Universidade Federal de São Paulo (UNIFESP), São Paulo 04023-062, Brazil; rosa.unifesp@gmail.com

**Keywords:** urinary tract infection, *Escherichia coli*, epidemiology, hybrid *E. coli* strains, DEC, UPEC, ExPEC, virulence factors, phylogeny, outpatients

## Abstract

Urinary tract infections (UTI) affect community and healthcare patients worldwide and may have different clinical outcomes. We assessed the phylogenetic origin, the presence of 43 virulence factors (VFs) of diarrheagenic and extraintestinal pathogenic *Escherichia coli*, and the occurrence of hybrid strains among *E. coli* isolates from 172 outpatients with different types of UTI. Isolates from phylogroup B2 (46%) prevailed, followed by phylogroups A (15.7%) and B1 (12.2%), with similar phylogenetic distribution in symptomatic and asymptomatic patients. The most frequent VFs according to their functional category were *fimA* (94.8%), *ompA* (83.1%), *ompT* (63.3%), *chuA* (57.6%), and *vat* (22%). Using published molecular criteria, 34.3% and 18.0% of the isolates showed intrinsic virulence and uropathogenic potential, respectively. Two strains carried the *eae* and *escV* genes and one the *aggR* gene, which classified them as hybrid strains. These hybrid strains interacted with renal and bladder cells, reinforcing their uropathogenic potential. The frequency of UPEC strains bearing a more pathogenic potential in the outpatients studied was smaller than reported in other regions. Our data contribute to deepening current knowledge about the mechanisms involved in UTI pathogenesis, especially among hybrid UPEC strains, as these could colonize the host’s intestine, leading to intestinal infections followed by UTI.

## 1. Introduction

Urinary tract infection (UTI) is the most prevalent bacterial infection worldwide and affects approximately 150 million people each year [1,2,3,4]. UTIs are frequent in community and healthcare infections and have different clinical outcomes, including cystitis, pyelonephritis, and asymptomatic bacteriuria [5,6]. These infections are caused by various microorganisms, including Gram-positive bacteria, Gram-negative bacteria, and fungi. However, the most common etiologic agent of community-acquired and healthcare-related infections is uropathogenic *Escherichia coli* (UPEC) [2,7].

*E. coli* is the main facultative anaerobe present in the intestine of humans and animals as an important member of their microbiota, participating in the maintenance and stability of intestinal homeostasis [8,9,10]. However, the genetic plasticity of the species allowed the emergence of pathogenic strains capable of causing intestinal and extraintestinal diseases [11,12,13].

About 50% of the *E. coli* genome comprises accessory genes acquired by horizontal gene transfer (HGT). The mixing of these genes in different combinations increases the capacity of *E. coli* to adapt to different niches and environments. Additionally, some of them contribute to their hosts’ colonization and infection processes [14,15,16,17].

The *E. coli* strains that produce diarrhea due to intestinal infections are known as diarrheagenic *E. coli* (DEC), which is divided into pathotypes, according to the combination of virulence factors (VFs) that determine virulence, colonization sites, and the signs and symptoms generated in the host. The DEC pathotypes are represented by: enteropathogenic *E. coli* (EPEC), Shiga toxin-producing *E. coli* (STEC), enterotoxigenic *E. coli* (ETEC), enteroaggregative *E. coli* (EAEC), enteroinvasive *E. coli* (EIEC), and diffusely adherent *E. coli* (DAEC) [18,19].

The *E. coli* strains that affect organs outside the intestine are known as extraintestinal pathogenic *E. coli* (ExPEC) [12,20]. Among the most common diseases that ExPEC can cause are UTI, sepsis, neonatal meningitis, and diseases that affect animals such as poultry, which are caused by UPEC, sepsis-associated *E. coli* (SEPEC), neonatal meningitis-associated *E. coli* (NMEC), and avian pathogenic *E. coli* (APEC), respectively [21,22,23,24].

These pathogens use adhesins, fimbriae, toxins, invasins, in addition to different escape mechanisms from the immune system to cause tissue damage and spread in the urinary tract [25]. These VFs contribute to the survival of UPEC in distinct niches and the pathogenicity of these microorganisms [25].

Since most genes related to *E. coli* virulence are inserted into mobile genetic elements that can be transferred to others by HGT, the mix of pathotypes might happen when one already established pathogenic strain acquires new genes from another pathotype [26].

Because of these findings, the term hybrid was adopted to describe the emergence of pathogenic *E. coli* strains carrying combinations of DEC and ExPEC diagnostic-related VFs or strains recovered from extraintestinal infections that harbor the DEC VFs used to characterize the various DEC pathotypes [15,26,27,28,29]. Although there are substantial studies that describe the epidemiology of these hybrid *E. coli* strains, most of them focus on the presence of hybrid strains in metropolitan or large cities or the screening of collections, and little is known about their frequency in smaller cities. To better address this topic, this study aimed to analyze the occurrence of hybrid strains of UPEC in isolates from outpatients with different types of UTI in the São Paulo countryside and evaluate their phenotypic and genotypic properties to determine the pathogenic potential of hybrid UPEC strains. Additionally, we provide information regarding the molecular epidemiology of UTI caused by *E. coli* in the community.

## 2. Materials and Methods

### 2.1. Patients and Bacterial Isolates

From July 2019 to December 2020, all *E. coli* strains isolated from urine culture with a bacterial counting of at least 10^5^ CFU/mL and identified by the BD Phoenix Automatic System were enrolled in this study.

The strains were from outpatients who attended a private clinical laboratory located in Mogi Guaçu, in the countryside of São Paulo state (Brazil), with urine culture demand. All information regarding host symptoms, gender, and age were obtained from anonymized medical records when it was available.

Using a collection and transport swab in Cary Blair gel medium (Cary Blair 132C, Venturi Transystem, COPAN, Murrieta, CA, USA), a single *E. coli* colony obtained from the urine culture was collected. The swabs containing the isolate were plated on MacConkey agar and incubated at 37 °C for 24 h. To confirm the *E. coli* species, after checking the purity, a single colony from each sample was evaluated by biochemical assays in EPM medium, MILi medium, and Simmon’s Citrate [30,31,32]. 

### 2.2. Phylogenetic Origin

The Clermont et al. [33] multiplex method to assign the strains into one of the eight phylogroups (A, B1, B2, C, D, E, F, or Clade I) was applied. Briefly, the *chuA*, *yjaA*, *arpA*, and TspE4.C2 genes were searched in a quadruplex PCR (Appendix A). A duplex PCR was performed if phylogroup identification was not possible in this first reaction. The PCR reactions were carried out using GoTaq^®^ Green Master Mix (Promega, Promega Corporation, Madison, WI, USA), and 1 µL of boiled bacterial suspension as template.

### 2.3. Virulence Genes Characterization

The presence of 43 virulence-associated genes was evaluated by PCR. Genes coding for the production of adhesins (*afaBC*III, *afaE*-VIII, *bfpB*, *bmaE*, *cf29A*, *daaE*, *eae*, *fimA*, *hra*, *iha*, *mat*, *papC*, *saa*, *sfaDE*, *shf*, *tsh*, *yfcV*), invasins (*invE*, *ompA*), protectins (*cvaC*, *kpsMT*II, *kpsMT*III, *ompT*, *traT*,), toxins (*cdtA*, *cnf1*, *eltA*, *estA*, *hlyA*, *pet*, *pic*, *sat*, *stx1*, *stx2*, *vat*), secretion system (*escV*), iron transport systems (*chuA*, *ireA*, *iroN*, *iucD*, *fyuA*, *sitA*), and the transcriptional regulator gene, *aggR*, were searched. The detailed conditions of each PCR reaction are described in Appendix A. 

The presence of specific molecular markers was used to classify *E. coli* into one of seven pathotypes as previously published [18,19,34,35] and summarized in Table 1. 

The molecular markers related to the identification of the intrinsic virulence (presence of two or more out of the *papC*, *sfaDE*, *afaBC*III, *iucD*, and *kpsMT*II genes) and uropathogenic potential (simultaneous presence of *vat*, *chuA*, *fyuA*, and *yfcV*) [34,35] were also searched to evaluate the frequency of strains harboring these traits among the strains studied (Table 1). 

The strains that harbored at least one of the molecular markers used for DEC pathotypes’ molecular diagnosis (Table 1) were considered hybrid strains [26] and were further evaluated for their phenotypic characteristics. 

### 2.4. Random Amplified Polymorphic DNA (RAPD)

The hybrid strains were analyzed by RAPD as previously described [36] to distinguish if they were highly related or different *E. coli* clones. This assay is based on two distinct PCR reactions, using a single primer each, 1247 (5′-AAGAGCCCGT-3′) and 1283 (5′-GCGATCCCCA-3′). The cycle conditions for the 1247 primer were: 95 °C for 5 min; 35 cycles (95 °C for 1 min, 38 °C for 1 min, 72 °C for 2 min); 72 °C for 10 min; and for the 1283 primer, they were: 95 °C for 5 min; 35 cycles (95 °C for 1 min, 36 °C for 1 min, 72 °C for 2 min); 72 °C for 10 min. Reactions were carried out as previously described.

The PCR products were then evaluated on the same agarose gel. The clonal evaluation was based on the amplification profile. If the strains presented the same amplification pattern in both PCRs, they were classified as clones.

### 2.5. Cell Culture, Maintenance and Adherence Assay

Assays using HeLa (ATCC^®^ CCL-2™), HEK 293T (ATCC^®^ CRL-1573™), and T24 (ATCC^®^ HTB-4™) cell lineages were performed. HeLa cells were used to determine the hybrid strains’ adherence patterns as usually performed in the diagnosis of typical (3 h) and atypical (6 h) DEC strains, as previously described [11,19]. The other cell lines, representing epithelial cells from the human bladder (T24 cells) and kidneys (HEK293T cells), were used to evaluate the ability of hybrid strains to colonize the urinary tract. The cultures maintenance for HeLa and HEK 293T cells was performed exactly as previously described [37]. T24 cells were maintained in McCoy 5A (modified) media (Sigma, Saint Louis, MO, USA), supplemented with 10% bovine fetal serum (BFS) (Gibco, Brazil), and 1% antibiotic mixture (penicillin—5 mg/mL, streptomycin—5 mg/mL; neomycin—10 mg/mL) (PSN) (Gibco, Carlsbad, CA, USA). Cells were maintained in 25 mL cell cultures flasks at 37 °C in an atmosphere of 5% CO_2._

The adherence assays were carried out using glass coverslips contained in 24-wells microplates at 80% cell confluence as previously described [38]. For the adherence assay, the cells were washed three times with sterile phosphate-buffered saline (PBS), and then, 1 mL of DMEM (for HeLa and HEK 293T) or McCoy 5A (for T24) with 2% of BFS were added in each well. In the HeLa cells assay, 2% methyl-D-mannose (Sigma-Aldrich, Saint Louis, MO, USA) was added to the medium to evaluate the adherence pattern. Each well was inoculated with 20 µL of each bacterial suspension containing 10^8^ CFU/mL and then, the microplates were incubated at 37 °C for 3 h or 6 h. Afterward, the cells were washed three times with PBS, fixed with methanol at room temperature for 30 min and stained with May-Grünwald/Giemsa (Merck, Darmstadt, Germany), diluted 1:1 and 1:2 in water, respectively. 

Bacterial adherence was evaluated under immersion light microscopy. The prototype strains EAEC 042 (aggregative adherence), EPEC E2348/69 (localized adherence), and aEPEC 4581-2 (localized adherence-like) were used as adherence patterns controls. *E. coli* HB101 and non-infected cells were used as negative controls, while the CFT073 strain was used as a UPEC control.

### 2.6. Biofilm Assay

A quantitative biofilm formation assay was performed with the hybrid strains as previously published [39]. Strains were grown in Lysogeny Broth (LB) at 37 °C for 18 h. After that, in 96-well polystyrene plates, 5 µL of each culture were added into 200 µL of LB or DMEM and incubated at 37 °C for 24 h. Afterward, the wells were gently washed with PBS, fixed with 3% formaldehyde, stained with 200 µL of 0.5% crystal violet, and 200 µL of 95% ethanol was added for dye solubilization. 

The wells were examined using a spectrophotometer (EnSpire Multimode Plate Reader, PerkinElmer, Waltham, MA, USA), at an optical density of 620 nm (OD_620_). The strains EAEC 042 and *E. coli* HB101 were used as positive and negative controls, respectively. Biofilm production was determined by comparing each strain with the non-biofilm producing strain, HB101. The strain CFT073 was used as a UPEC control, and non-infected wells were used as dye retention control. Results were obtained from the average of an experiment conducted in triplicate.

### 2.7. Statistical Analyses 

GraphPadPrism 5.0 (GraphPad Prism Software, Inc., San Diego, CA, USA) was used to perform the analyses. The One-way ANOVA followed by post hoc Tukey HSD test was used to compare the results, and the threshold for statistical significance was *p*-value ≤ 0.05. 

## 3. Results

### 3.1. Epidemiological Data on Infections Caused by UPEC Strains

A total of 172 *E. coli* isolates was assessed, of which 143 (83.1%) were from female patients with a mean age of 49 years, ranging between 4 and 94 years and median of 54.5 years; and 7 (4%) male patients with a mean age of 62 years, ranging between 44 and 80 years and median 60.5 years. Information related to gender and age was unavailable for 22 patients (22.7%). 

The type of infection was determined for 131 patients (76.1%), in which 37 (28%) were asymptomatic bacteriuria (ABU) and 94 (71.7%) were symptomatic UTI, including 17 (12.9%) recurrent infections. 

### 3.2. Classification of ExPEC Virulence Profile, Uropathogenicity, and Phylogenetic Origin of Hybrid UPEC Strains

The 172 urine isolates were analyzed for phylogenetic origin, presence of DEC and ExPEC markers, genetic classification of ExPEC virulence, and uropathogenic potential, as described above. Regarding the phylogenetic origin of the strains, 46% belonged to phylogroup B2, 15.7% to phylogroup A, 12.2% to phylogroup B1, 8.1% to phylogroup C, 2.3% to D, 5.2% to E, 2.9% to F, and 7.5% of the strains were not identified by the methodology used (Table 2).

Among the 43 genes searched, 14 have been described in DEC isolates (including the pathotype diagnosis markers), 16 described in ExPEC, and 13 described in both pathotypes. Thirty genes were found, and it was observed that the most frequent VFs according to their functional category were *fimA* (94.8%), *ompA* (83.1%), *ompT* (63.3%), *chuA* (57.6%), and *vat* (22%) representing adhesins, invasins, protectins, iron uptake systems, and toxins, respectively. Virulence genes common to DEC strains but unrelated to pathotype definition were found (*saa*, *pet*, *shf*, *cdtA*, and *daaE*), with the STEC autoagglutinating adhesin gene *saa* being the most frequent (37 isolates, 21.5%). All VFs studied are presented in Table 2.

Considering the molecular classification for intrinsic virulence (ExPEC+) and uropathogenic potential (UPEC+), 34.3% (59 strains) were classified as ExPEC+ and 18% (31 strains) as UPEC+. Together, ExPEC+ and UPEC+ sum 71 (41.3%) strains, including 19 *E. coli* strains that had virulence markers compatible with both definitions (Table 2). 

Evaluation of the distribution of VFs considering the molecular classification showed that 15 VFs were statistically related to the ExPEC+ and UPEC+ classifications, including the *saa* and *pet* genes, which are commonly related to DEC pathotypes (Table 2). The *ireA*, *pic*, and *sat* genes were exclusively associated with ExPEC+ strains, while *kpsMT*III and *cdtA* were related to UPEC+ strains. 

Additionally, the phylogroup B2 was related to ExPEC+ and UPEC+ strains, whereas phylogroups A and B1 were related to those strains that were negative for both classifications. Three hybrid pathogenic strains were found in this study (1.7%), two of which presented the *eae* gene and lacked the *bfpB* gene (LSC 073 and LSC 183). For this reason, they were provisionally classified as UPEC/aEPEC. The third hybrid strain harbored the *aggR* gene, characteristic of EAEC, being provisionally classified as UPEC/EAEC (LSC 052). The three hybrid strains comprised different clones as detected by RAPD (not shown) and their different phylogenetic origin (phylogroups B2, B1, and D, respectively) (Appendix A).

It was not possible to identify any difference between symptomatic and asymptomatic patients regarding the phylogenetic origin of the strains or the presence of molecular markers related to ExPEC+ and UPEC+ molecular classification (*p* > 0.05) (Appendix A). The identified hybrid isolates were isolated from symptomatic and asymptomatic patients (Appendix A).

The VFs profile, phylogenetic origin, and the characterization of intrinsic virulence and uropathogenic potential of the strains studied are described in Appendix A. 

The diagnostic marker genes of the other DEC pathotypes researched (ETEC, EIEC, and STEC) were not identified among the studied isolates.

### 3.3. Adherence Pattern of Hybrid UPEC Strains

The interaction assays on HeLa cells were performed within 3 h of incubation for the UPEC/EAEC strains and within 6 h for the UPEC/aEPEC strains. The latter strains adhered poorly in 3-h assays, which is an expected behavior observed in aEPEC strains [11,19]. In the two UPEC/aEPEC strains, the formation of small loose clusters of bacteria adhered to the cell surface that discern the localized adherence-like pattern (LAL) was observed, which is characteristic of aEPEC (Figure 1). The UPEC/EAEC strain showed an aggregative adherence pattern characteristic of typical EAEC.

### 3.4. Interaction with HEK 293T Cells

HEK 293T cells were used in this study to assess the adherence capacity of hybrid UPEC strains in the urinary tract. The UPEC/aEPEC strains were able to adhere to these cells and were randomly distributed on the coverslips, and the UPEC/EAEC strain was able to adhere firmly to this cell lineage (Figure 2).

### 3.5. Interaction with T24 Cells

The T24 lineage was used to analyze the adherence capacity of hybrid UPEC strains in the urinary tract. The UPEC/aEPEC strains, despite being able to adhere, did so with less intensity and more dispersed and the UPEC/EAEC strain was able to adhere very well to this cell lineage (Figure 3).

### 3.6. Biofilm Formation Assays

Biofilm formation assays revealed that the hybrid UPEC/aEPEC and UPEC/EAEC strains in this study were unable to produce biofilm in the media and conditions tested (not shown). 

## 4. Discussion

*E. coli* is currently the primary etiological agent of UTI [3,7,40,41,42], being responsible for about 80% of the cases, demonstrating the importance of studies on the genetic characteristics of this species [43,44,45]. However, most studies regarding molecular epidemiology of the VFs presented by *E. coli* strains isolated from bacteriuria were carried out in metropolitan regions or large cities, and epidemiological data from the countryside are barely known. In the present work, we assessed the phylogenetic origin, VFs, and the occurrence of hybrid strains among *E. coli* isolates from bacteriuria of outpatients in Mogi Guaçu, a city located in Sao Paulo state countryside bearing 153,033 inhabitants. Our results showed that most of the UPEC isolates identified in this study were recovered from female patients. This finding corroborates with other previous studies that report women of all ages as the most affected patients [46], mainly due to the anatomy of the female urinary tract that favors the emergence of these infections [47].

When analyzing the phylogenetic origin of the isolates, we found that phylogroup B2 was predominant in the study population. This finding is supported by other studies that report the predominance of phylogroup B2 in UPEC isolates, such as those published by Paniagua-Contreras et al. [48], Rezatofighi et al. [49], and Lin et al. [50] that showed the prevalence of 51%, 55%, and 65.7% of UTI in females, respectively. Interestingly, it is worth noting that in the present study, strains belonging to phylogroups A and B1 represent about 27% of the isolates. These phylogroups are known to generally harbor commensal *E. coli* strains; however, especially in recent years, some studies have demonstrated ExPEC strains belonging to these phylogroups [37,51,52,53,54], and our data contribute effectively to corroborate their importance in community-acquired UTI in the symptomatic population evaluated. The phylogenetic distribution was similar in symptomatic and asymptomatic patients, with phylogroup B2 being the most frequent, followed by phylogroups A and B1 for both types of patients. 

A great diversity of adhesin-encoding genes was observed among the virulence genes analyzed, with *fimA*, which encodes type 1 fimbria, being the most frequent. Adhesive structures such as fimbriae and pili are essential VFs, as it is through these structures that the bacterial adhesion process to host cells occurs, which is essential for the initial stages of the infection [55]. Other studies conducted in different regions of the world also found a high prevalence of genes encoding type 1 fimbriae [46,48,55,56,57]. Virulence genes such as *papC*, *sfaDE*, *ompT*, and *kpsMT*II found in the isolates of the present study have been frequently reported in UPEC strains worldwide [49,50,56].

Among the UPEC isolates studied, we also detected genes that are involved in iron uptake systems (*sitA*, *chuA*, *iucD*, *fyuA*, *iroN,* and *ireA*) [55,58,59]. Interestingly, 63.9% of the isolates had the *traT* and/or *kpsMT* genes. These genes are related to the capacity of the bacteria to resist the bactericidal activities of the human serum and neutrophils. These mechanisms are an important advantage to pathogenic bacteria that ascend the urinary tract, reach the kidneys, and cause pyelonephritis, which is the main source of *E. coli* bloodstream infections. Additionally, the *traT* gene is plasmid located and associated with the plasmid conjugative-transfer apparatus. Therefore, it means that about 42% of the evaluated strains harbor conjugative plasmids that may harbor genes encoding VFs, antimicrobial resistance, or both. This high frequency is problematic since, in many cases, bloodstream infections originate from strains that can access the bloodstream through the kidneys. It is believed that this access is facilitated by the P fimbriae [60,61], which were detected in 24.5% of our isolates. Previous studies have also found a high frequency of *traT*, with frequencies varying between 42% and 73.2% [48,56].

To assess the strains that were considered potentially more pathogenic, we used the criteria proposed by Johnson et al. [34] and Spurbeck et al. [35] to characterize the intrinsic virulence and the uropathogenic potential here referred to as ExPEC+ and UPEC+. Interestingly, our data showed that 34.3% of the isolates were classified as ExPEC+ and 18.0% as UPEC+. Although these markers strongly indicate pathogenic potential, they are not sufficient to identify all *E. coli* strains recovered from extraintestinal infections of symptomatic outpatients, especially in the studied population. Additionally, we did not find a difference in the frequency of these markers comparing host symptomatology. Moreover, other studies reported the occurrence of ExPEC devoid of these classic virulence markers, some of them in strains with epidemiological relevance [20,62,63,64,65,66]. Few studies evaluated the significance of the UPEC+ classification in strains isolated from bacteriuria [67,68,69,70]. Still, the frequency found in the present study is by far the lowest reported since the UPEC+ frequency reported in the referred studies was above 50%. Moreover, studies that searched for UPEC+ among human bloodstream isolates reported frequencies above 44% [71,72,73]. However, it is remarkable that these studies were carried out in North America and Europe, which may suggest a link with the geographic region evaluated.

Nevertheless, strains classified as ExPEC+ or UPEC+ bear more VFs than those that do not meet the criteria, and 20 VFs were significantly more frequent in these types of strains. Fifteen of these VFs were shared by both groups, including *hlyA*, *hra*, *iroN*, *ompT*, *pet*, *saa*, *traT*, and the VFs related to the ExPEC+ and UPEC+ molecular classification (*vat*, *chuA*, *fyuA*, *yfcV*, *papC*, *sfaDE*, *iucD*, and *kpsMT*II). Five VFs were exclusively associated with one of the groups, with *ireA*, *sat*, and *pic* being associated with the ExPEC+ classification and *kpsMT*III and *cdtA* being associated with UPEC+. 

Regarding the VFs searched, we found in the UPEC strains studied eight VFs that are commonly related to the DEC pathotypes. Genes like *saa*, *pet*, *shf*, *cdtA*, and *daaE* were rarely assessed in this type of strain. Among them, *saa* and *pet* called the attention by their association with the ExPEC+ and UPEC+ classification. The *saa* gene encodes the STEC autoagglutinating adhesin, reported in ExPEC strains in a frequency lower than that found in the present study (21.5%) [74]. Interestingly, it was previously reported in a STEC strain isolated from UTI. The *pet* gene codes for a toxin, which belongs to the Serine protease autotransporters of Enterobacteriaceae (SPATE) family and was associated with EAEC strains. This toxin was previously reported in ExPEC strains isolated from different sources. The frequency found here (9.9%) was higher than those found among strains isolated from bloodstream infections in adults and newborns with meningitis [75,76], similar to the frequency found in one study that evaluated strains from UTI [77], and lower than another study that assessed strains isolated from bacteremia of children with diarrhea [78].

There have been increasingly frequent reports of infections caused by hybrid *E. coli* strains in recent years. This term has been used to describe pathogenic *E. coli* strains carrying new combinations of DEC and ExPEC VFs or strains recovered from extraintestinal infections that had DEC VFs [15,26,27,28,29]. Currently, several studies have reported the occurrence of hybrid pathogens in diverse countries around the world; in fact, the occurrence of this type of pathogen was already reported in all continents [27,51,54,74,79,80,81,82,83,84].

Given the importance of these hybrid strains, we investigated the presence of DEC markers in our collection, and the results showed the presence of two strains that carried the *eae* and *escV* genes and one with the *aggR* gene, which classified them as UPEC/aEPEC and UPEC/EAEC, respectively. The presence of the ***eae*** and *aggR* markers in UPEC strains have been previously described, revealing what would be an increased potential in the virulence arsenal of these strains, as they could have the capacity to cause intestinal and extraintestinal infections [26,51,74]. A previous study developed by Valiatti et al. [54] with an aEPEC/UPEC strain reinforces this hypothesis, as it demonstrated the functionality of the LEE region in causing the attaching/effacing lesion in HeLa cells. However, in our study, the two UPEC/aEPEC strains found were isolated from ABU, while the UPEC/EAEC strain was isolated from one symptomatic patient. These conflicting data reinforce the necessity for additional studies evaluating hybrid strains and the host symptoms to increase the knowledge regarding the pathogenic potential of all hybrid strains. Overall, the frequency of hybrid strains in UPEC collections in Sao Paulo is low. In two previous studies from our group developed by Abe et al. [85] and Nascimento et al. [37], this frequency was 3% and 2%, respectively, and in both, the highest frequency observed was related to UPEC/EAEC strains, unlike the data we present here, where we saw a higher number of UPEC/aEPEC strains. However, in some countries like Mexico, UPEC/EAEC occurs in a much higher frequency, 22% [86], and similar frequencies were reported in Mozambique, but in strains isolated from bacteremia in children [78]. Interestingly, in our collection and the studies mentioned above, the presence of UPEC/STEC strains was not observed, which has been reported in other studies [74,87] showing that, apparently, this type of hybrid strain is not yet circulating in Sao Paulo.

The three hybrid strains in our study were evaluated for their ability to interact in vitro with HeLa and urinary tract (HEK 293T and T24) cells. The UPEC/aEPEC strains showed a LAL pattern that is commonly observed in aEPEC strains, and the UPEC/EAEC strain showed the AA pattern. Previous studies with UPEC/aEPEC isolates corroborate our findings, as these also showed a LAL pattern [37,54]. Interestingly, Nascimento et al. [37] found that when analyzing the adherence pattern of seven UPEC/EAEC isolates, five did not show the typical AA pattern. We also found that all strains were able to interact with HEK 293T and T24 cell lines, which reinforces the uropathogenic potential of these strains. Furthermore, the hybrid strains were unable to produce biofilm under the conditions tested. Generally, biofilm-producing strains are associated with greater severity and/or persistence of the infection [88]; however, in general, the ability of UPEC strains to form biofilm varies [69,89,90,91,92,93]. So far, the few studies that have evaluated biofilm formation by hybrid strains have demonstrated that this property is also variable among these strains [37,54,86].

## 5. Conclusions

Finally, the data presented here provide an in-depth and detailed analysis of UPEC strains, particularly hybrid strains, which have genotypic markers of DEC and ExPEC. We believe that these data are of great interest and may contribute to deepening current knowledge about the mechanisms involved in the pathogenesis of urinary tract infections, especially among hybrid UPEC strains, as these could colonize the host’s intestine, leading to intestinal infections followed by UTI.

## Figures and Tables

**Figure 1 microorganisms-10-00302-f001:**
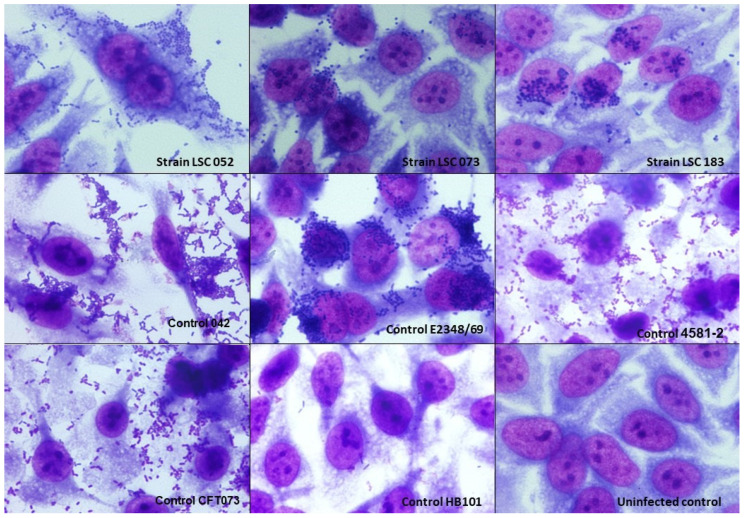
Adherence patterns of hybrid uropathogenic *Escherichia coli* (UPEC) strains to HeLa cells. Adherence patterns were evaluated in assays with incubation periods of 3 h or 6 h, at 37 °C, in the presence of 2% D-mannose, using a multiplicity of infection of 10. The preparations were stained with May-Grünwald/Giemsa and observed under an optical microscope (magnification ×1000). All hybrid UPEC strains were adherent. Strain LSC 052, showing the aggregative adherence (AA) pattern; strains LSC 073 and LSC 183, showing the localized adherence-like (LAL) pattern. *E. coli* strains used as controls: 042 (prototype enteroaggregative *E. coli* (EAEC) expressing AA; E2348/69 (prototype typical enteropathogenic *E. coli* (EPEC) expressing a localized adherence (LA) pattern); 4581-2 (atypical EPEC (aEPEC) showing the LAL pattern); CFT073 (prototype UPEC strain); HB101 (*E. coli* K-12-derived laboratory strain, non-adherent).

**Figure 2 microorganisms-10-00302-f002:**
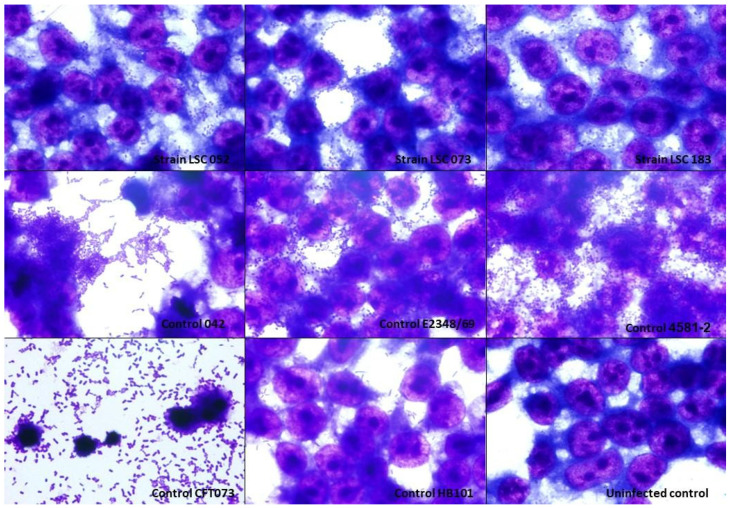
Interaction of hybrid uropathogenic *Escherichia coli* (UPEC) strains with HEK 293T cells of renal origin. The ability of hybrid UPEC strains to interact with human kidney cells was evaluated in assays with an incubation period of 3 h, at 37 °C without D-mannose, using a multiplicity of infection of 10. The preparations were stained with May-Grünwald/Giemsa and observed under an optical microscope (magnification ×1000). All hybrid UPEC strains interacted with the kidney cells in different strengths. *E. coli* strains used as controls: 042 (prototype EAEC); E2348/69 (prototype typical EPEC); 4581-2 (atypical EPEC); CFT073 (prototype UPEC); HB101 strain (non-adherent *E. coli* K-12-derived laboratory strain).

**Figure 3 microorganisms-10-00302-f003:**
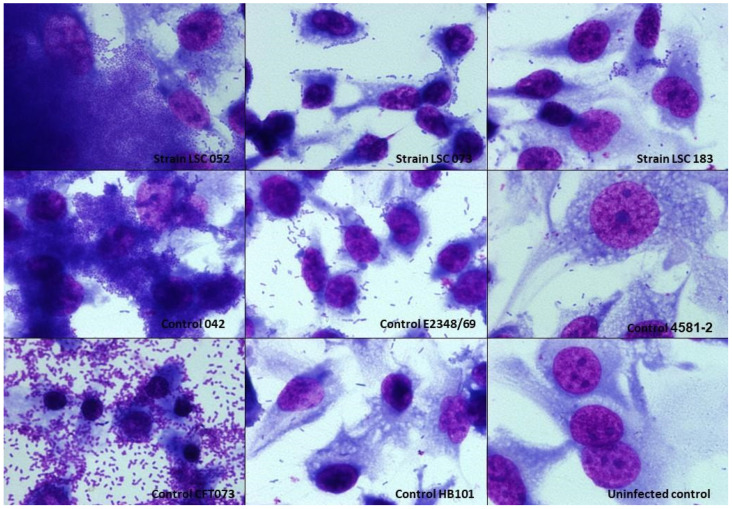
Interaction of hybrid uropathogenic *Escherichia coli* (UPEC) strains with T24 cells of bladder origin. The ability of hybrid UPEC strains to interact with human bladder cells was evaluated in assays with an incubation period of 3 h, at 37 °C, without D-mannose, using a multiplicity of infection of 10. The preparations were stained with May-Grünwald/Giemsa and observed under light microscopy (magnification ×1000). Hybrid UPEC/enteroaggregative *E. coli* (EAEC) strain (LSC 052) and hybrid UPEC/atypical enteropathogenic *E. coli* (aEPEC) strains (LSC 073 and LSC 183) interacted with bladder cells at different intensities. *E. coli* strains used as controls: 042 (prototype EAEC); E2348/69 (prototype typical EPEC); 4581-2 (atypical EPEC); CFT073 (prototype UPEC); HB101 strain (non-adherent *E. coli* K-12-derived laboratory strain).

**Table 1 microorganisms-10-00302-t001:** Molecular markers used to identify the *Escherichia coli* pathotypes.

Pathotype	Molecular Markers
EAEC	*aggR*
ETEC	*elt* and/or *est*
EIEC	*invE*
typical EPEC	*bfpB*, *eae*, *escV*
atypical EPEC	*eae*, *escV*
STEC	*stx1* and/or *stx2*
EHEC	*stx1* and/or *stx2*, *eae*, *escV*
ExPEC+ (intrinsic virulence) ^a^	*≥*2 among *papC*, *sfaDE*, *afaBCIII*, *iucD*, *kpsMTII*
UPEC+ (uropathogenic potential) ^a^	*vat*, *chuA*, *fyuA*, and *yfcV*

^a^ The *Escherichia coli* strains isolated from UTI were considered to belong to the ExPEC/UPEC pathotype. The usage of the molecular markers intends to identify strains harboring a more specific pathogenic profile as previously described [26,34,35].

**Table 2 microorganisms-10-00302-t002:** Frequency of DEC and ExPEC virulence factors (VFs) and phylogroups of urinary tract infection strains (*n* (%)).

		All Isolates	ExPEC+ ^a^	UPEC+ ^b^	Non-UPEC+/ExPEC+
Traits	VFs	*n* = 172	*n* = 59	*n* = 31	*n* = 101
**Adhesins /Invasins**	*fimA*	163 (94.8)	57 (96.6)	29 (93.5)	95 (94)
	*ompA*	143 (83.1)	46 (77.9)	27 (87.1)	86 (85.1)
	*yfcV*	72 (41.8)	36 (61) ***	31 (100) ***	24 (23.7)
	*mat*	72 (41.8)	30 (50.8)	13 (41.9)	37 (36.6)
	*hra*	61 (35.4)	32 (54.2) **	17 (54.8) *	26 (25.7)
	*papC*	48 (27.9)	40 (67.8) ***	14 (45.2) ***	7 (6.9)
	*saa*	37 (21.5)	23 (38.9) ***	12 (38.7) **	10 (9.9)
	*sfaDE*	28 (16.2)	23 (38.9) ***	8 (25.8) *	5 (4.9)
	*iha*	19 (11)	7 (11.8)	4 (12.9)	10 (9.9)
	*shf*	14 (8.1)	2 (3.3)	2 (6.4)	10 (9.9)
	*daaE*	2 (1.1)	2 (3.3)	0 (0)	0 (0)
	*eae*	2 (1.1)	0 (0)	1 (3.2)	1 (0.9)
	*tsh*	1 (0.5)	0 (0)	1 (3.2)	0 (0)
**Protectins**	*ompT*	109 (63.3)	52 (88.1) ***	28 (90.3) ***	46 (45.5)
	*kpsMT*II	77 (44.7)	51 (86.4) ***	24 (77.4) ***	20 (19.8)
	*traT*	72 (41.9)	34 (57.6) *	20 (64.5) *	32 (31.6)
	*kpsMT*III	2 (1.1)	1 (1.6)	2 (6.4) *	0 (0)
	*cvaC*	16 (9.3)	8 (13.6)	5 (16.1)	7 (6.9)
**Iron acquisition systems**	*chuA*	99 (57.6)	53 (89.8) ***	31 (100) ***	34 (33.6)
	*fyuA*	89 (51.7)	37 (62.7) *	31 (100) ***	40 (39.6)
	*sitA*	88 (51.1)	35 (59.3)	20 (64.5)	46 (45.5)
	*iroN*	57 (33.1)	33 (55.9) ***	21 (67.7) ***	18 (17.8)
	*iucD*	41 (23.8)	29 (49.1) ***	9 (29) *	11 (10.9)
	*ireA*	15 (8.7)	8 (13.6) *	4 (12.9)	4 (3.9)
**Toxins**	*vat*	38 (22)	19 (32.2) ***	31 (100) ***	7 (6.9)
	*sat*	22 (12.7)	12 (20.3) *	4 (12.9)	8 (7.9)
	*hlyA*	18 (10.4)	16 (27.1) ***	5 (16.1) *	2 (1.9)
	*pet*	17 (9.9)	11 (18.6) *	7 (22.6) **	3 (2.9)
	*pic*	9 (5.2)	7 (11.8) *	2 (6.4)	1 (0.9)
	*cdtA*	6 (3.5)	4 (6.7)	4 (12.9) *	1 (0.9)
**Transcriptional regulator**	*aggR*	1 (0.5)	0 (0)	0 (0)	1 (0.9)
**Secretion system**	*escV*	2 (1.1)	0 (0)	1 (3.2)	1 (0.9)
**ExPEC+**		59 (34.3)	N.A.	19 (61.2)	N.A.
**UPEC+**		31 (18)	19 (32.2)	N.A.	N.A.
**Hybrid pathogenic *E. coli* ^c^**		3 (1.7)	0 (0)	1 (3.2)	2 (1.9)
**Phylogroups**					
**A**		27 (15.7)	0 (0) ^(***)^	0 (0) ^(**)^	27 (26.7)
**B1**		21 (12.2)	2 (3.3) ^(*)^	0 (0) ^(*)^	19 (18.8)
**B2**		79 (46)	49 (83) ***	30 (96.8) ***	19 (18.8)
**C**		14 (8.1)	4 (6.7)	0 (0)	10 (9.9)
**D**		4 (2.3)	2 (3.3)	0 (0)	2 (1.9)
**E**		9 (5.2)	2 (3.3)	0 (0)	7 (6.9)
**F**		5 (2.9)	0 (0)	0 (0)	5 (4.9)
**NC**		13 (7.5)	0 (0) ^(*)^	1 (3.2)	12 (11.8)

^a^ intrinsic virulence as determined by the presence of at least two among the genes *papC*, *sfaDE*, *afaBCIII*, *iucD*, and *kpsMT*II; ^b^ uropathogenic potential, as determined by the simultaneous presence of *vat*, *chuA*, *fyuA*, and *yfcV*; ^c^ strains that harbor a molecular marker related to DEC diagnosis (*eae*, *escV*, or *aggR*). N.A., not applicable. The differences in the frequency of VFs present in strains classified as ExPEC+ or UPEC+ were compared with those present in strains negative for both classifications, here named as non-ExPEC+/UPEC+ strains. Fisher exact test, * *p* ≤ 0.05; ** *p* < 0.001; *** *p* < 0.0001. The parenthesis means that the trait is negatively related with the ExPEC+/UPEC+ classification or statistically related to non-UPEC+/ExPEC+ group. All strains were negative to the following VFs: *afaBC*III, *afaE*-VIII, *bfpB*, *bmaE*, *cf29A*, *invE*, *cnf1*, *eltA*, *estA*, *stx1*, and *stx2*.

## Data Availability

Not applicable.

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
