# Peer review of "Molecular Epidemiology and Presence of Hybrid Pathogenic Escherichia coli among Isolates from Community-Acquired Urinary Tract Infection"

_microorganisms, 2022, doi:10.3390/microorganisms10020302_

Round 1

Reviewer 1 Report

After reading the manuscript entitled "Molecular epidemiology and presence of hybrid pathogenic Escherichia coli among isolates from community-acquired urinary tract infection", I consider it appropriate to be published in Molecules.

The subject matter is original and important. The introduction section presents background information, this part is concise and well‐written. The materials and Methods section is written correctly. Results and Discussion sections are written in an interesting way. The table and figures are a valuable supplement to the text. The changes in the text must be performed to increase its quality.

 Affiliation

Affiliations should be written in English.

 Materials and Methods

page 4 is: … ethanol were added…, should be: … ethanol was added…

Results

page 5 is: … 7.1% to D, 2% to E…, should be: … 1.7% to D, 5.2% to E… - such data are in Table 2

page 5 in the text is: … fimA (94.7%)…, in table 2 is fimA 94.8% - choose the correct value

page 7 Could you explain why the interaction assays on HeLa cells were performed within 3 h of incubation for the UPEC/EAEC strains and within 6 h for the UPEC/ aEPEC strains?

pages 7-8   Why three different cell lines (HeLa, HEK 293T, and T24) were used in the research? Please explain.

page 9 On what basis was there no biofilm production found? What criterion was used? It seems quite strange that among 172 E. coli strains, none produced biofilm.

Discussion

page 9 is: … prevalence…, should be: … the prevalence…

page 10 is: … representing about 27%…, should be: … represent about 27%…

page 10 is: … represented only 3%…, should be: … representing only 3%…

page 10 is: … source for E. coli…, should be: … source of E. coli

page 10 is: … in this type of strains…, should be: … in these types of strains. …

Author Response

We acknowledge the useful comments and suggestions.

Reviewer 2 Report

In this manuscript, Nascimento et al characterized 3 hybrid E. coli strains isolated from patients with UTI. The topic is interesting and the manuscript is well-organized. I have only one comment as described below,

Material and Method 2.4. The author described that they performed RAPD to distinguish the clonality of E. coli strains. However, they did not showed the data. In addition, RAPD could be used to determine whether examined strains are "highly related strains" but not "identical strains".   

Author Response

(The authors gave the same response as above.)

Round 2

Reviewer 2 Report

The authors have addressed all my concerns and therefore I support publication of this manuscript.

Author Response

No modifications were requested by this reviewer.

We revised the manuscript details in this version, which were carefully checked, including language, italic of genes and species, and references order. The modifications are shown as tracking corrections. In addition, we have completed the legends of Table 2 and Figures 2 and 3 to conform with the Microorganisms instructions. We also inserted information on two genes (iroN and escV) in Table 2 (and the discussion section), which we found missing. 

Please, note that we have not removed the phrases in yellow because they are the modifications we had introduced to respond to the reviewers' comments in the previous version. We did not modify these paragraphs in this current version.

We appreciate the opportunity to correct these mistakes and hope that this version is adequate to be accepted.

All authors agree with the modifications introduced to this final version of our manuscript.

Sincerely yours,

Tânia Gomes, PhD
